# Inequalities in birth before arrival at hospital in South West England: a multimethods study of neonatal hypothermia and emergency medical services call-handler advice

Laura Goodwin [1], Kim Kirby [1,2] Graham McClelland,[3] Emily Beach,[1] Adam Bedson,[4] Jonathan Richard Benger,[1] Toity Deave,[5] Ria Osborne,[6] Helen McAdam,[7] Roisin McKeon-Carter,[8] Nick Miller,[1] Hazel Taylor,[9] Sarah Voss [1]

For numbered affiliations see end of article.

**Correspondence to**
Dr Kim Kirby;
kim.kirby@uwe.ac.uk

## ABSTRACT

**Objectives** To examine inequalities in birth before arrival (BBA) at hospitals in South West England, understand which groups are most likely to experience BBA and how this relates to hypothermia and outcomes (phase A). To investigate opportunities to improve temperature management advice given by emergency medical services (EMS) call-handlers during emergency calls regarding BBA in the UK (phase B).

**Design** A two-phase multimethod study. Phase A analysed anonymised data from hospital neonatal records between January 2018 and January 2021. Phase B analysed anonymised EMS call transcripts, followed by focus groups with National Health Service (NHS) staff and patients.

**Setting** Six Hospital Trusts in South West England and two EMS providers (ambulance services) in South West and North East England.

**Participants** 18 multidisciplinary NHS staff and 22 members of the public who had experienced BBA in the UK.

**Results** 35% (64/184) of babies conveyed to hospital were hypothermic on arrival. When compared with national data on all births in the South West, we found higher percentages of women with documented safeguarding concerns at booking, previous live births and 'late bookers' (booking their pregnancy >13 weeks gestation). These women may, therefore, be more likely to experience BBA. Preterm babies, babies to first-time mothers and babies born to mothers with disability or safeguarding concerns at booking were more likely to be hypothermic following BBA. Five main themes emerged from qualitative data on call-handler advice: (1) importance placed on neonatal temperature; (2) advice on where the baby should be placed following birth; (3) advice on how to keep the baby warm; (4) timing of temperature management advice and (5) clarity and priority of instructions.

**Conclusions** Findings identified factors associated with BBA and neonatal hypothermia following BBA. Improvements to EMS call-handler advice could reduce the number of babies arriving at hospital hypothermic.

## STRENGTHS AND LIMITATIONS OF THIS STUDY

⇒ To the best of our knowledge, this is the first study to investigate the temperature of babies arriving at hospital following a prehospital birth in South West England.

⇒ This is the first study to investigate neonatal temperature management advice provided by emergency service call-handling triage systems.

⇒ The service evaluation of hospital neonatal records had a small sample size and was confined to one geographical area which limits the representativeness of our findings.

⇒ Focus groups represent the views of a small sample of participants and this may limit the transferability of the findings.

⇒ Our research was conducted using data that spanned the COVID-19 pandemic when UK maternity services were significantly impacted and this may have influenced our findings.

## BACKGROUND

In the UK, an estimated 3700 births a year are unplanned out-of-hospital births or births before arrival at hospital (BBA).[1 2] These births are attended by ambulance staff such as paramedics, emergency care assistants and ambulance technicians, usually without the attendance of a qualified midwife or obstetrician. BBA is associated with unfavourable perinatal outcomes and increased mortality.[3–5]

An important risk factor for mortality following BBA is hypothermia,[5–7] which is defined by the WHO as a temperature below 36.5°C.[8] The mortality risk from hypothermia is especially prominent in premature babies; in low birth weight infants, mortality increases by 28% per 1°C decrease in admission temperature below 36.5°C.[9] Paramedics are usually the first on

scene at BBAs, however, recent audits of emergency medical service (EMS) data in the UK suggest that paramedics do not routinely take newborn temperatures (temperatures recorded in 2%–10% of cases).[10 11] When temperatures have been recorded, the majority have indicated neonatal hypothermia (72%).[10 11] Currently, there are no published UK data to show the percentage of babies who are hypothermic on arrival to hospital following a BBA and the resulting outcomes for these babies.

To prevent neonatal hypothermia and poor outcomes associated with neonatal hypothermia following BBA, it is important to understand which women are most likely to experience BBA, which babies are most likely to be cold on arrival, and how this interacts with outcomes such as mortality, length of hospital stay, and the need for special care and interventions. Literature around inequalities in maternal and perinatal outcomes suggests an increased risk of poor outcomes for women who do not engage or have little engagement with maternity services.[2 12] This includes concealed or teenage pregnancies, as well as women from migrant or minority ethnic backgrounds.[13 14] A call to EMS regarding labour or birth may, therefore, be the first point of contact that some women will have regarding their pregnancy.

In the prehospital setting, babies can become hypothermic within minutes[15] and are sometimes born before EMS arrival. In these instances, the newborn is more likely to be hypothermic on arrival at hospital, potentially due to ineffective warming by non-professionals.[16] Therefore, the earlier that effective warming measures can be initiated (eg, at the point of calls to EMS), the better the chances of improved outcomes following BBA. A recent Healthcare Safety Investigation Branch (HSIB) report[17] highlighted the fact that UK investigations into poor patient outcomes following BBA do not routinely consider the clinical impact of advice given by EMS call-handlers or recommended work with relevant stakeholders to develop guidance for maternity emergencies during these calls. It is necessary to investigate the advice given regarding neonatal temperature management during EMS calls relating to BBA and work with relevant stakeholders to investigate potential ways to develop and improve this advice.[17]

The aims of this project were as follows:

1. To examine inequalities in BBA in South West England and understand which groups are most likely to experience BBA, and also how this relates to hypothermia and neonatal outcomes (phase A).
2. To investigate opportunities to improve the temperature management advice given by EMS call-handlers during emergency calls regarding BBA in the UK (phase B).

## METHODS
### Setting
Hospital data were collected from six National Health Service (NHS) Hospital Trusts in South West England. The area covered by these Trusts has one of the lowest population densities in the country, with people living across a mix of coastal communities, isolated rural areas and urban centres.[18]

Data for EMS calls were collected from two NHS ambulance service Trusts using the two different Clinical Decision Support Systems used in the UK: the Advanced Medical Priority Dispatch System (AMPDS) (Salt Lake City, USA) and NHS Pathways (NHS Digital, Leeds, UK). Data from calls triaged by AMPDS were collected from the South Western Ambulance Service NHS Foundation Trust, while data from calls triaged by NHS Pathways were collected from the North East Ambulance Service NHS Foundation Trust. AMPDS has been in use in the USA since 1978 and is used by emergency services in 59 countries, including roughly half of ambulance service Trusts in the UK.[17] NHS Pathways is used by the other UK ambulance trusts and is overseen by a National Clinical Governance Group, which is hosted by the Royal College of General Practitioners and is made up of representatives from UK medical royal colleges.[17]

### Patient and public involvement
A dedicated patient and public involvement (PPI) group guided this work from its inception, supporting the development of research questions and study design. The group consisted of four women and two men who had experience of BBA. Experiences were varied and represented ethnic, cultural and geographical diversity. Over the course of the project, the PPI group reviewed and refined participant-facing documents for stakeholder focus groups, contributed to data synthesis and advised the team on the approach to recruitment and dissemination.

### Study design
This multimethods study consisted of both qualitative and quantitative data with two phases. Phase A included an analysis of hospital data and phase B included content analysis of EMS calls, followed by thematic analysis of focus groups with NHS staff and patients.

The two phases of this study were funded as one piece of work and linked through their focus on inequality. While we wanted to use phase A of this work to examine potential inequalities surrounding BBA births in the South West of England, we also wanted to examine the accessibility of EMS call-handler advice regarding neonatal hypothermia. This would help to identify potential opportunities to improve this advice by ensuring we were aware of the populations most at risk and providing the EMS call-handlers with accessible and easily understandable information.

### Phase A: neonatal hypothermia on arrival at hospital
Anonymised extracts from routinely collected, non-identifiable data (hospital neonatal records for BBA babies), were provided by participating Trusts for a 3-year period (January 2018–January 2021). These were compared with National Maternity Data from the same geographical area o understand which groups are most

likely to experience BBA and how this would relate to hypothermia and neonatal outcomes recorded by Hospital Trusts (including admission to a neonatal unit (NNU), length of hospital stay and treatment provided).

## Recruitment

Hospital Trusts were eligible to take part if they were covered by the South West Academic Health Science Network (AHSN).[18] The South West AHSN presented information about the study in a number of online events and invited Trusts to email an expression of interest to the study lead to take part. Participating Trusts were offered administration costs to collect and anonymise data.

## Data collection

An Excel spreadsheet (Microsoft, Redmond, USA) was created to standardise the information provided by Hospital Trusts. Data points selected for inclusion were all demographic characteristics for the mother and baby, and all care and outcome information related to neonatal temperature, which were available from the maternity notes held by the Hospital Trusts. The spreadsheet was piloted and refined by midwives and neonatal nurses at one of the participating Trusts, before circulating. Members of maternity and neonatal staff at each participating Trust searched their records for cases of BBA, uploaded anonymised information to the spreadsheet and returned it to the study lead. Records were included if they related to a live birth in the prehospital setting where the baby was brought to hospital by EMS (ambulance staff).

Due to the small number of BBA cases recorded over the time period specified above, we did not do an in-depth statistical analysis with statistical testing. As such, demographic characteristics of the mothers (booking status, parity, safeguarding status) and characteristics of the birth (gestation, temperature on admission, treatment) are reported using descriptive statistics.

## Phase B: 999 call-handler advice on neonatal temperature management

### Analysis of current advice

Two UK NHS EMS provider organisations (Ambulance Trusts) provided anonymised audiorecordings of calls relating to BBA that were analysed to investigate the temperature management advice given by EMS call-handlers during emergency calls.

### Data collection

All EMS calls relating to BBA between January 2021 and January 2022 were searched by the clinical information and records (CIR) teams at the two participating EMS provider organisations (ambulance services). The team used an EMS diagnosis code of 'normal delivery' (a single live birth with no obstetric complications) to find relevant calls. Calls were listened to by the CIR team to check

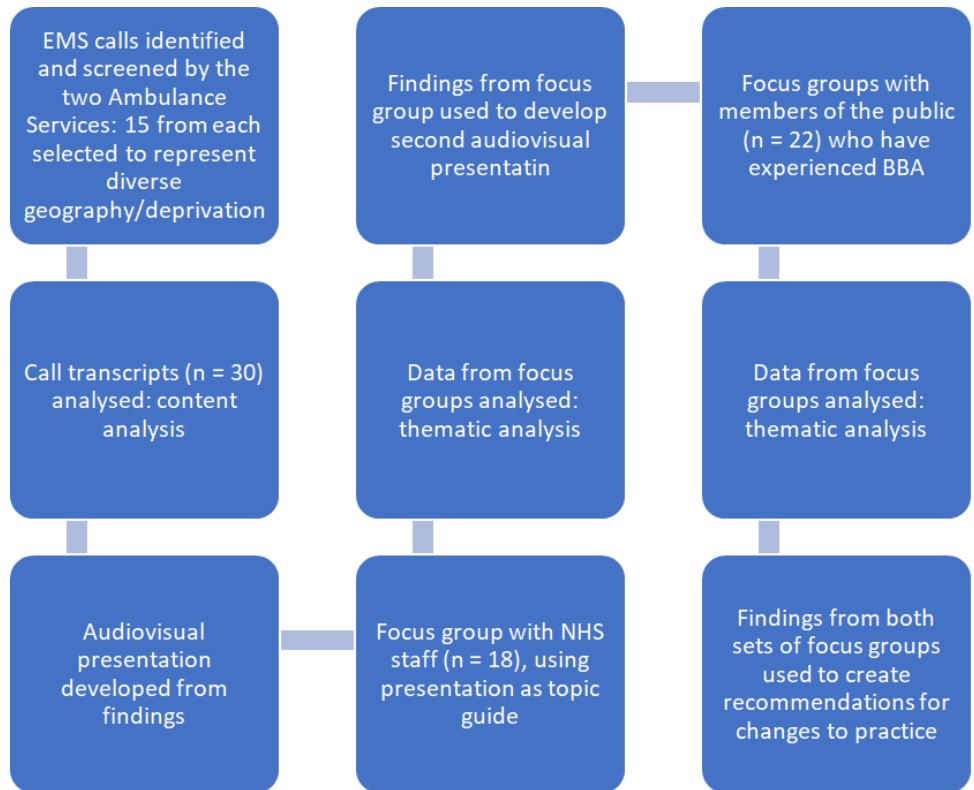

**Figure 1** Flow chart to show the process of how call-handler data and focus group data were identified, screened and analysed. EMS, emergency medical services.

eligibility, and 30 eligible calls (15 from each ambulance service) were selected at random from postcodes with varying levels of deprivation (figure 1). Deprivation was determined by the team through the lower layer super output area of the incidents, which gave an overall index of multiple deprivation score for each call. Anonymised audiorecordings of the calls were then transferred to the study lead and transcribed by a university-approved transcription company.

Data were analysed by LG and AJ using content analysis, and findings were used to develop an audiovisual presentation to stimulate discussion during focus groups (figure 1).

## Staff and patient views

Focus groups were conducted with NHS staff and patients to investigate opportunities to improve the temperature management advice given during EMS calls.

## Recruitment

A purposive sampling technique was adopted to ensure rich information about the topic.[19] Participants were eligible to take part in staff focus groups if they were a practising member of NHS staff with experience of BBA (eg, paramedic, midwife, neonatal nurse/doctor, EMS call-handler). Participants were eligible to take part in patient focus groups if they had experienced BBA (eg, as a parent or caller to EMS).

Recruitment adverts were circulated using social media and collaborating organisations (South West AHSN, Stillbirth and Neonatal Death Society charity), and participants were reimbursed with a £25 e-voucher for their time. Eligible participants were invited to contact LG to join a focus group. Potential participants were emailed a participant information sheet, privacy notice and consent form. Verbal confirmation of consent was audiorecorded at the start of each focus group. We aimed to recruit 30–40 participants, guided by information power with a narrow aim and specific study population.[20]

## Data collection

Four focus groups were held via Teams (Microsoft, Redmond, USA) with NHS staff, facilitated by LG. The content and structure of BBA calls were discussed in relation to their ideas on best practice temperature management advice, in response to findings from the content analysis of EMS calls, using the presentation described above (figure 1).

Themes identified during these focus groups were used to create a second audiovisual presentation to facilitate focus group discussions with members of the public (figure 1). Five focus groups were held via teams and focused on the content and accessibility of the advice given during EMS calls, as well as the suggestions put forward by NHS staff.

Focus groups were videorecorded and transcripts were generated by teams. Transcripts were coded and analysed by LG using thematic analysis. Findings were synthesised to create a set of recommended adaptations to the current advice given by call-handlers using EMS triage systems. These recommended adaptations were sent to focus group participants and the PPI group for final review and refinement.

# RESULTS

## Phase A: neonatal hypothermia on arrival at hospital

217 babies were conveyed to the 6 hospital Trusts by the ambulance service during the above time period (live births ≥24 weeks). There were 33 records (15%) with no neonatal arrival temperature documented.

Of those with a temperature recorded, 35% (64/184) were hypothermic on arrival at hospital. Table 1 gives a breakdown of these recorded temperatures according to WHO classifications of core body temperature for newborns.[8]

## Inequalities in chances of having a BBA

Hospital data from this study were compared with South West data from the National Maternity Data Set for 2021–2022.[21]

Some characteristics were more prevalent in the data for BBAs compared with local data for all births (table 2). This included safeguarding concerns documented at booking, previous live births and late booking (past 13 weeks). A definition of all characteristics is provided in online supplemental file 1.

While we set out to compare the ethnicity of women experiencing BBA with the overall South West data, this

**Table 1** Neonatal temperature on arrival at hospital

| WHO classification | Temperature range | Number of babies (%) n=184 | Implication for outcomes/care (WHO) |
|---|---|---|---|
| Hyperthermia | 37.6°C | 1 (1) | Cause for concern |
| Normal range | 36.5°C–37.5°C | 119 (65) | N/A—normal care |
| Cold stress (mild hypothermia) | 36°C–36.4°C | 48 (26) | Cause for concern |
| Moderate hypothermia | 32°C–35.9°C | 14 (8) | Danger; warm baby |
| Severe hypothermia | <32°C | 2 (1) | Outlook grave, skilled care urgently needed |

**Table 2** Comparisons made between BBA data and overall data for the South West

| Characteristic | BBA data Number (%) | Overall south west data Number (%) |
|---|---|---|
| Safeguarding concerns documented at booking | n=217 | n=48 900 |
| Yes | 47 (22) | 4180 (8) |
| No | 170 (78) | 43 425 (89) |
| Missing data | 0 (0) | 1295 (3) |
| Previous live birth(s) | n=217 | n=48 905 |
| Yes | 185 (85) | 26 875 (55) |
| No | 32 (15) | 19 125 (39) |
| Missing data | 0 (0) | 2905 (6) |
| Gestation at booking* | n=217 | n=48 900 |
| Early booker (up to 12+6 weeks) | 179 (83) | 46 020 (94) |
| Delayed booker (13 to 19+6 weeks) | 20 (9) | 1835 (4) |
| Late booker/not booked (≥20 weeks) | 14 (6) | 935 (2) |
| Missing data | 4 (2) | 110 (0.2) |
| Mother's age at booking | n=217 | n=48 900 |
| Under 20 | 7 (3) | 1410 (3) |
| 20–29 | 90 (41) | 20 000 (41) |
| 30–39 | 108 (50) | 25 725 (53) |
| 40 or over | 11 (5) | 1765 (4) |
| Missing data | 1 (0.1) | 0 (0) |

*Booking classifications are presented as defined by the National Institute for Health and Care Excellence.[30]
BBA, birth before arrival.

was prevented by the use of different ethnicity categories in the two data sets. For those experiencing BBA recorded ethnicity was: British European 94%, East European 2%, other 2%, Indian 0.5%, North European 0.5%, South African 0.5% and South European 0.5%. For those in the South West data set ethnicity was white 84%, Asian or Asian British 3%, black or black British 2%, mixed 2% and any other ethnic group 2%. We were also unable to consider social deprivation, as we had hoped to do, as we were only provided with the first half of patients' postcodes by the Trust. This was done to maintain anonymity of patient data.

### Inequalities in neonatal hypothermia/poor outcomes

Hypothermia on arrival at hospital was more prevalent for preterm babies, babies born to mothers who had not had a previous live birth and babies born to mothers who reported a disability or safeguarding concern at booking (table 3).

Babies with neonatal hypothermia on arrival at hospital were more likely to be admitted to an NNU or receive advanced care (33% vs 19%). Advanced care was classified as any care needed in addition to 'normal newborn care', for example, the provision of oxygen or intravenous medication.[22]

Of the hypothermic babies needing advanced care/admission to an NNU, a slight majority were born at term (11/21; 52%), one had a diagnosis of sepsis confirmed (5%) and just over half (11/21; 52%) were either moderately or severely hypothermic on arrival.

Seven (11%) hypothermic babies had an extended length of stay in hospital (>7 days) whereas none of the warm babies did. Due to the small sample size we were unable to perform in-depth statistical analysis to control for potential confounding variables. However, most (6/7, 86%) of the babies with an extended length of stay were also preterm. More treatment was given to hypothermic babies (glucose: 8% vs 1%; radiant heat source: 20% vs 3%; and incubator: 11% vs 3%).

### Phase B: EMS call-handler advice on neonatal temperature management

#### Participant characteristics

18 NHS staff took part in the four staff focus groups. Most participants were female (13/18, 72%). Participants' job titles comprised midwives, paramedics, call-handlers, learning development officer, neonatal nurses, neonatal doctors and advanced care practitioners.

22 members of the public who had experience of prehospital birth took part in 5 focus groups. Most participants were female (18/22, 82%). Participants had a range of experiences, representing diverse geographical regions, areas of deprivation and prehospital birth experience (eg, at home, in the car and outside).

Five main themes were identified as affecting neonatal temperature management during EMS calls. These included (1) the importance placed on neonatal temperature; (2) advice on where the baby should be placed following birth; (3) advice on how to keep the baby warm; (4) the timing of temperature management advice and (5) the clarity and priority of instructions. Illustrative quotes are provided in table 4.

#### Theme 1: the importance placed on neonatal temperature

Although call-handlers from both triage systems repeated the instruction to 'keep mum and baby warm' at several points during the EMS call, participants noted that the importance of neonatal temperature was not explicitly addressed. This was of concern to participants who had experienced a BBA, who said that they were unaware of how quickly babies could become hypothermic in the prehospital setting, or how this could affect neonatal outcomes. It was noted by participants that antenatal information did not provide much information on neonatal hypothermia so they were unprepared for it (Q1).

Most participants reported that call-handlers had focused their advice and instructions on the person giving birth (eg, determining whether the placenta had been delivered/whether there was any bleeding), and

Table 3  Characteristics of mothers and babies, by neonatal temperature on arrival at hospital, n=184

| Characteristic of baby/mother | | Hypothermia <36.5°C n=64 Number (%) | Normothermia 36.5°C–37.5°C n=119 Number (%) |
|---|---|---|---|
| Prematurity | Preterm (<37 weeks) | 9 (14) | 2 (2) |
| | Term (37+weeks) | 54 (84) | 116 (97) |
| | Unknown | 1 (2) | 1 (1) |
| Previous live birth | Yes | 48 (75) | 109 (92) |
| | No | 16 (25) | 10 (8) |
| Disability reported at booking | Yes | 6 (9) | 2 (2) |
| | No | 58 (91) | 115 (97) |
| | Unknown | 0 (0) | 2 (2) |
| Safeguarding concerns reported at booking | Yes | 19 (30) | 20 (17) |
| | No | 45 (70) | 99 (83) |
| Gestation at booking | Early booker (up to 12+6 weeks) | 51 (80) | 101 (85) |
| | Delayed booker (13 to 19+6 weeks) | 6 (9) | 13 (11) |
| | Late booker/unbooked (≥20 weeks) | 6 (9) | 3 (2) |
| | Unknown | 1 (2) | 2 (2) |
| Mother's ethnicity | British European | 58 (91) | 114 (9%) |
| | East European | 3 (4.5) | 0 (0) |
| | Indian | 0 (0) | 1 (1) |
| | North European | 1 (1.5) | 0 (0) |
| | South African | 1 (1.5) | 0 (0) |
| | South European | 0 (0) | 1 (1) |
| | Other | 1 (1.5) | 4 (3) |
| Mother born in UK | Yes | 57 (89) | 112 (94) |
| | No | 6 (9) | 5 (4) |
| | Unknown | 1 (2) | 2 (2) |
| Model of care | Consultant led | 25 (39) | 48 (40) |
| | Midwifery led | 37 (57) | 70 (59) |
| | Unknown | 2 (3) | 1 (1) |
| Mother's attendance at antenatal appointments | All attended | 52 (81) | 88 (74) |
| | Not all attended | 12 (19%) | 25 (21) |
| | Unknown | 0 (0) | 6 (5) |
| Characteristic of baby/mother | | <36.5°C n=64 Median (IQR) | ≥36.5°C n=119 Median IQR |
| Mother's age at booking | | 28 (23–34)) | 31 (27–35)) |

that limited attention had been given to the baby once it was established that they were breathing. Some reported that their baby had become hypothermic following their BBA and expressed frustration that it had not been made a priority or been communicated how important it was to keep their baby warm (Q2).

All participants felt that the importance of temperature could be further highlighted. Participants who had experienced prehospital birth suggested that this should be communicated in clear and simple language. NHS staff

suggested that giving the caller the 'job' of keeping the baby warm could help emphasise this importance and increase the chance of temperature management being performed well (Q3).

### Theme 2: where the baby should be placed following birth

The two triage systems differed in their instruction on where the baby should be placed following the birth. The AMPDS triage system states that callers should 'keep the baby between the mother's legs and level with her

**Table 4** Illustrative quotes from qualitative data from focus groups

| Quote no. | Focus group | Quote |
|---|---|---|
| 1 | Public focus group 2 | Nothing was mentioned about keeping the baby warm and to be honest, I had no idea that that's something that I should be doing. |
| 2 | Public focus group 2 | When I delivered my daughter, their concern was mainly on delivering the placenta. It was my mum that had suggested that I should take my baby into my jumper that I was wearing to keep her warm because it was a cold morning. It was only until they realised that they couldn't deliver my placenta that they told me to get the blanket out and then wrap my daughter. |
| 3 | Staff focus group 2 | You'll find that people want something to do. They want to focus in on something. And if you were to give that information around the importance of thermal regulation 'keeping that baby warm is really important now/ When that baby's born, we really want to keep that baby warm.' I think that would be really useful for that person who's trying to do something positive. |
| 4 | Staff focus group 2 | Often when we're reviewing cases, we'll kind of note that as we kind of arrived on scene, the baby is on the floor and the mother's just in shock and the babies on a cold floor and it's drafty. |
| 5 | Public focus group 4 | What we found really really confusing is that all the way through we were advised skin-to-skin, and don't do anything with the cord - keep that connected as long as possible. But then when it came to the actual 999 caller, it was totally different advice and it was 'keep the baby on the back'…Having that totally conflicting advice put my partner in the position where he was like 'I'm totally confused'. |
| 6 | Public focus group 4 | When the baby did come out, she was just lying on the bed on her back. And I instinctively wanted to take her and put her on my chest like I did before with the midwives, and the instruction was to leave her on her back. I wasn't allowed to pick her up. |
| 7 | Public focus group 2 | I was in a shopping mall, and there were lots of people around me. Lots of people looking and crowding. So I couldn't feel like I could do skin-to-skin. I was in panic and the security guy was a man. I'm sad I didn't get to do it, but it was too much stress. |
| 8 | Public focus group 4 | I completely did not think straight in that moment at all…I was just a bit of an idiot, to be honest. And so when he came out, I didn't think about keeping him warm…I don't think you can rely on people thinking straight in that moment. … I think sometimes like I said, stating the obvious is quite useful in that kind of scenario. |
| 9 | Public focus group 4 | We had quite a long wait before the midwife came. My husband was really panicking…but actually if someone had said to him 'move to a warm room' or 'go and put the heating on, if you haven't got it on already', he would have done that straight away. |
| 10 | Staff focus group 1 | The gap should be used to paraphrase back to them. Just saying, you know…like 'dry, warm and wrapped'. Just keeping it simple repeating it back to them, showing that they understand the instruction for…however long it takes (the paramedics) to finally get there. |
| 11 | Public focus group 4 | If it seems like it's a rushed labour, then perhaps maybe focus on what's going on at hand. And then perhaps maybe give the (temperature) advice thereafter like step by step… Whereas if it's maybe not as quick, maybe strike some sort of conversation with the person that's with the mother and say 'ok, when the baby does come out, remember to keep the baby warm. |
| 12 | Public focus group 1 | If they can [provide warming advice] straight away if they've got time, then yeah, do it again also in between [gaps in the conversation]. If they've got time waiting for the ambulance, just mention it again. Just maybe check, check baby's chest or, you know, make sure it's warm or just little things like that because you do, you do forget. |
| 13 | Staff focus group 4 | Babies being born in the sac is really rare. Usually it's you know, contractions, suddenly the water's gone, and the baby follows it quite quickly with these mums…So I don't think that should even be in there really because that might even worry mums like 'well, should there be a sac? Is it wrong that they're not in a sac? Oh my goodness me, you know, where's the sac?' And then start looking for a sac. |

bottom', while NHS Pathways states that callers should 'place the baby on the mother's chest or tummy'.

Participants expressed frustration at the instruction to place the baby between the mother's legs and felt this could be very detrimental to temperature management, especially if the mother was in a room with a cold floor, such as a bathroom or kitchen (Q4).

Participants who had received this advice following prehospital birth said that it had caused confusion, as it contradicted what they had been taught antenatally (ie, skin-to-skin contact following birth) (Q5).

For those who had given birth in a car, this advice had proven especially challenging. One participant noted that by trying to keep the baby between her legs her husband

had had to open all the car doors, which had potentially contributed to her baby being hypothermic on arrival at hospital.

Some participants spoke of not being 'allowed' to pick their baby up following the BBA and to keep their baby on its back on the bed/floor until the paramedics arrived (Q6).

Skin-to-skin contact with the mother was put forward by NHS staff as the optimal practice for neonatal normothermia following prehospital birth. There was some discussion among staff regarding the importance of maintaining a clear airway during skin-to-skin contact, and it was acknowledged that regular prompts would need to be given by call-handlers to check that the baby's nose and mouth could be seen. Participants also emphasised the need for clear and simple instructions to be given by call-handlers on how to conduct skin-to-skin contact effectively, including checking to see if the mother was warm themselves.

Most participants said that skin-to-skin contact would have been practical in the circumstances surrounding their BBA. Some circumstances were seen as potentially problematic for skin-to-skin contact, for example, if the cord was too short, they were too anxious, or were not confident in handling the baby (eg, first-time parents). Some participants expressed concerns over privacy regarding skin-to-skin contact and said that they would not feel confident doing this in a public place or if they thought male paramedics might attend (Q7).

Staff also acknowledged that there may be cases where skin-to-skin contact was not appropriate or possible, for which staff suggested thorough drying and wrapping of the baby before placing them on someone warm.

Ensuring that the baby was put in the optimal place was discussed by participants as particularly important in the context of long wait times for ambulances and the 'cost of living crisis'. Participants noted that people could potentially be waiting for hours for an ambulance in a cold house, so ensuring the baby is kept as warm as possible should be a priority when advising callers where to place the baby following birth.

### Theme 3: how to keep the baby warm

Neither of the two triage systems contain specific advice on how to keep the baby warm, although some call-handlers did add some advice themselves during calls (eg, suggesting the use of a hat). Generic advice of 'keep mum and baby warm' was given by each pathway, and callers were also told to 'cover the baby's head, but not its face' and to 'use towels and blankets to keep the baby warm'.

During focus groups with NHS staff, participants expressed a preference for warming advice to be more specific and targeted. Suggestions included advising skin-to-skin contact (as above), putting a hat on the baby, considering possible changes to the environment (eg, closing any windows, moving off cold floors, putting the heating on), and emphasising the need to dry the baby thoroughly and replace wet towels with dry ones.

Participants also suggested adding an instruction for callers to check the warmth of the baby by touching the skin on the chest/back, to see if warming measures were working.

The specific advice put forward by staff was viewed positively in focus groups with those who had experienced a BBA. Many suggested that they had experienced stress and anxiety during their BBA, and that this had caused them to 'lose all common sense' or be unable to think rationally. As such, the instruction to 'keep the baby warm' was viewed as too generic and ambiguous by participants, who felt that clear, practical instructions would be easier to follow and would help them to feel more in control of the situation (Q8).

Most participants felt that all warming measures suggested by staff would have been practical in the circumstances of their BBA, and that alternatives would have been easily considered if not (eg, using an item of clothing instead of a blanket/towel). Participants did acknowledge, however, that some warming measures may not be practical in all situations, and that callers should be advised that each warming measure should be carried out 'if possible'. Participants also noted that listing all warming measures may not be appropriate if the woman was calling for herself and had no-one around to assist her.

Of the measures suggested by staff, potential changes to the environment were emphasised by participants as something that could have considerably influenced their baby's temperature. Participants suggested that simple advice such as moving to a warmer room would have given them a sense that they were positively impacting the situation and helped relieve some of the anxiety and panic (Q9).

### Theme 4: timing of temperature management advice

Another key theme about the content and structure of the EMS triage systems was the timing of neonatal temperature management advice. While call-handlers from both pathways focused this advice predominantly after the baby was born, long gaps were identified in some calls while call-handlers waited for a part of the baby to be seen by the caller. NHS staff suggested that these gaps could be used to emphasise the importance of temperature, and to give advice on warming the environment if appropriate. Gaps in the conversation were also identified during periods of time where call-handlers had completed the triage algorithm and were waiting for paramedics to arrive. These gaps were often filled with small-talk. Staff suggested that this would be an appropriate time to give specific warming advice, and for call-handlers to check that essential warming measures had been completed (Q10).

Participants who had experienced a BBA felt that warming advice should be given as soon as possible but without interrupting vital instructions. Some participants felt that the woman should be the priority and focus of calls before the baby was born, but that a brief mention

of the importance of temperature would be helpful, and that focus should then shift to keeping the baby warm after birth (Q11).

Participants agreed with NHS staff that once the triage algorithm was complete, any additional time spent on calls waiting for the ambulance should be used to provide specific warming advice and to check it had been followed appropriately. Participants felt that it would be helpful to be reminded of all the steps they could take to keep their baby warm at any possible opportunity (Q12).

Focus group participants also discussed advice that could be given antenatally to prepare them for neonatal temperature management in the prehospital setting. Suggestions included an antenatal leaflet on the importance of neonatal thermoregulation, a reminder at antenatal appointments, and/or advice to pack a blanket in their hospital bag (in case of birthing in the car).

### Theme 5: clarity and priority of instructions

The clarity and priority given to instructions were discussed by participants in both focus groups. Several instructions were highlighted by participants as potentially confusing or distracting and some felt that the timing and priority given to these instructions could delay neonatal temperature management. For example, calls from the NHS Pathways algorithm included instructions for the caller to 'wipe or peel away the sac' before drying and wrapping the baby. This frequently caused confusion to callers, as none of the babies were born in their sac and meant that drying and wrapping instructions were delayed while the call-handler and caller discussed the nature of the sac. NHS staff suggested that the previous instruction to 'wipe the baby's nose and mouth' should be sufficient to cover any issues with the sac and suggested that this instruction be removed or altered to avoid confusion. Participants from both sets of focus groups suggested that once the baby was born, only essential instructions should be given before warming advice, as to avoid any confusion or distraction (Q13).

Other instructions questioned by NHS staff included the advice to get towels and blankets to 'protect the bed or floor', and the caller being directed to wash their hands once a part of the baby could be seen.

## DISCUSSION

This is the first study, to our knowledge, to investigate the temperatures of babies arriving at hospital following a prehospital birth (BBA) in South West England. We found that 35% of babies born prehospital arrived at hospital hypothermic (<36.5°C), and 25% of these were classed as 'moderately' or 'severely' hypothermic, with warming needed urgently to prevent poor outcomes. These findings are in line with those of a study in Finland, which found hospital diagnosis of neonatal hypothermia to be recorded in 53% of babies born out of hospital, compared with only 0.7% of babies born in hospital.[4] Our findings also concur with research from France, which found that 56% of babies born outside of hospital were hypothermic on arrival at hospital,[16] and that neonatal hypothermia was associated with increased rates of NICU admission.[5] Although our findings suggest better rates of neonatal normothermia following BBA than literature from other areas, 35% remains a significant proportion of babies arriving at hospital hypothermic. This could potentially be improved through better availability of optimal equipment such as transwarmers and accurate temperature monitoring devices, further training, increased awareness among ambulance staff[10] and improvements in EMS call-handler advice.

We found that particular babies may be more at risk of hypothermia on hospital arrival following BBA. These include preterm babies, babies of first-time mums, babies of women where there are safeguarding concerns and babies of women who reported disabilities at their booking appointment. While prematurity and low birth weight are established risk factors for hypothermia in newborns generally,[23 24] there is a paucity of previous literature exploring the other potential risk factors we identified. It may be that where BBA occurs these factors make it harder for ambulance staff to effectively maintain the temperature of the newborn. This may be due to a lack of maternal awareness of the risks of hypothermia prior to EMS arrival and the nature of the birthing environment which may be unprepared for delivery. The findings from this study suggest that there is a lack of consistent advice provided to EMS callers prior to the arrival of the ambulance that does not effectively support temperature management at birth. While our study is limited by the small sample size and focused geographical location these preliminary associations suggest hypotheses and provide the basis for future research, as well as helping NHS services to identify babies who may require additional warming and support following a BBA. Further work should replicate and expand our study to consider areas of high population density, with a larger sample size to enable a more detailed statistical analysis.

Our findings suggest that there may be some women in South West England who are more likely to experience BBA, which previous research suggests may result in poorer outcomes for their babies.[4 5] These include women with safeguarding concerns, women who book their pregnancies with a health professional later than 13 weeks gestation, and those who have had previous babies. Multiparity, late booking and/or lack of antenatal care are known to be associated with BBA from previous literature,[2 4 12 25] however, safeguarding concerns at booking have not been identified previously. Safeguarding concerns might include denial of pregnancy and maternal substance misuse. It seems likely that these factors are associated with delays in identifying the onset of labour and presenting to maternity services. It is also well known that the average duration of labour in multiparous women is shorter than it is in nulliparous women.[26] Due to the small sample size of this study, caution should be exercised when interpreting results, but this finding

may help maternity services identify and support women who are more likely to experience BBA. Despite receiving information on mothers' ethnicity from our BBA data, we were unable to explore any associations, as local ethnicity data were recorded differently from the National Maternity Data Set.[21] Work is ongoing to resolve this issue.

This study is the first to investigate neonatal temperature management advice provided by EMS call-handling triage systems. EMS dispatching is complex, and a call-handler must triage the call to inform the prioritisation of resource dispatch while also providing first aid advice prior to the arrival of an EMS resource (prearrival advice).[27] Our research identifies issues with unclear and contradictory prearrival advice during UK EMS calls regarding BBA and suggests five themes that could influence neonatal thermal care during these calls. NHS staff and members of the public made several recommendations for changes to the script to address potential barriers to good thermal care, based on these themes. These changes have already been implemented in NHS Pathways, and similar changes are pending approval in AMPDS.

Although this is the first study to investigate neonatal temperature management advice given by EMS call-handlers, increasing attention is being paid to call-handler advice and interactions with patients,[28] especially during maternity cases.[17] Indeed, the HSIB report into 'Maternity prearrival instructions by 999 call handlers'[17] similarly found that the two triage systems in England provide different prearrival instructions for birth, associated with different implications and risk. There is a growing body of evidence investigating the call-handler and caller interaction during EMS calls,[28] and this research should be considered in any proposed changes to emergency maternity prearrival advice. While much of the research into call-handler and caller interaction during the EMS call has focused on the out-of-hospital cardiac arrest (OHCA) scenario, these findings can be used to inform emergency maternity prearrival instructions and align with our own findings. For example, simplified instructions have been associated with a shorter interval between instruction and first chest compression in telephone cardiopulmonary resuscitation instruction in OHCA.[29] Future research could usefully investigate the design, implementation and impact of emergency maternity prearrival advice.

## CONCLUSION

The findings of this study suggest there may be certain women who are more likely to have a BBA in South West England and certain babies who are more likely to be hypothermic on arrival at hospital following a BBA. Findings also suggest potential areas for improvement to EMS call-handler advice regarding neonatal temperature management. Future work in this area has the potential to reduce the number of babies arriving at hospital hypothermic and improve longer-term outcomes, thereby addressing health inequalities by disproportionately benefiting underserved groups.

**Author affiliations**
[1]School of Health and Social Wellbeing, University of the West of England, Bristol, UK
[2]Research, Audit and Quality Improvement Department, South Western Ambulance Service NHS Foundation Trust, Exeter, UK
[3]Northumbria University, Newcastle upon Tyne, UK
[4]South Western Ambulance Service NHS Foundation Trust, Exeter, UK
[5]Centre for Child and Adolescent Health, University of the West of England, Bristol, UK
[6]Research, Audit & Improvement, South Western Ambulance Service NHS Foundation Trust, Exeter, UK
[7]Health Sciences and Wellbeing, University of Sunderland, Sunderland, UK
[8]University Hospitals Plymouth NHS Trust, Plymouth, UK
[9]Research Design Service, University Hospitals Bristol NHS Foundation Trust, Bristol, UK

**Acknowledgements** We would like to thank the NHS staff and members of the public who took part in focus groups for this evaluation, as well as the NHS Hospital and Ambulance Service Trusts who provided data. We would also like to thank: Alice Jennings from UWE Bristol, for supporting the collection of hospital data and contributing to analysis of EMS calls; Sally Hedge from the South West AHSN for supporting and promoting this work; and the members of our Patient and Public Involvement Group for their invaluable input and contributions to this study: Tasha Cooper, Farzana Kausir, Poonam Barnes, Louise Walker, Simon Walker and Arif Hoque. Finally, we would like to thank the funder of this project, the South West AHSN, and the study sponsor, UWE Bristol. This work was carried out by members of the Research in Emergency Care, Avon Collaborative Hub (REACH Bristol).

**Contributors** LG was the chief investigator/guarantor with overall responsibility for the study, and participated in study conception, design and coordination, performed and coded the focus groups, analysed the data and drafted the manuscript. HT participated in study conception, design and interpretation of the results, and supported quantitative data analysis. GM, KK, EB, AB, TD, RO, HM, RM-C, NM, SV and JRB participated in study conception, design and interpretation of the results. All authors were responsible for the critical revision of the manuscript for publication and approved the final version to be published.

**Funding** The study was funded by the South West Academic Health Science Network (AHSN) through its Perinatal Equity programme.

**Competing interests** None declared.

**Patient and public involvement** Patients and/or the public were involved in the design, or conduct, or reporting, or dissemination plans of this research. Refer to the Methods section for further details.

**Patient consent for publication** Not applicable.

**Ethics approval** This study involves human participants and ethics committee approval for this service evaluation was granted by the University of the West of England (UWE Bristol) Faculty of Health and Applied Sciences Research Ethics Committee (HAS.22.01.061). Participants gave informed consent to participate in the study before taking part.

**Provenance and peer review** Not commissioned; externally peer reviewed.

**Data availability statement** Data are available on reasonable request.

**ORCID iDs**
Laura Goodwin http://orcid.org/0000-0002-9118-4620

Kim Kirby http://orcid.org/0000-0002-8092-7978
Sarah Voss http://orcid.org/0000-0001-5044-5145

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
