## [Reviewer comments · BMJ Open]

ARTICLE DETAILS

TITLE (PROVISIONAL)	Inequalities in birth before arrival at hospital in South West England: a multi-methods study of neonatal hypothermia and Emergency Medical Services call-handler advice
AUTHORS	Goodwin, Laura; Kirby, Kim; McClelland, Graham; Beach, Emily; Bedson, Adam; Bengler, Jonathan; Deave, Toity; Osborne, Ria; McAdam, Helen; McKeon-Carter, Roisin; Miller, Nick; Taylor, Hazel; Voss, Sarah

VERSION 1 – REVIEW

REVIEWER	Bellini, Carlo Giannina Gaslini Institute
REVIEW RETURNED	06-Nov-2023

GENERAL COMMENTS	I read the article Manuscript ID bmjopen-2023-081106 very carefully. General impression. The topic is certainly important, but I would also say that in a certain sense it is obvious. It is not difficult to imagine that a newborn born outside a well-organized hospital could have greater risks of complications, including hypothermia. Having said this, there are some unclear things that prevent me from tackling the review work in detail. First look. The authors state that in the UK, approximately 3,700 births per year occur outside of hospital. Frankly, that seems like a huge number to me. I admit that I know approximately the birth data in the UK, but I would say that the percentage of births outside hospital is therefore around 1.2-1.3% (Wales, Scotland, England) approximately the same as Iceland and Switzerland. In Europe is decidedly below 1% and even in Italy we are at 0.05-0.06%. Assuming that these numbers are correct, but obviously I expect the authors to comment on them, another decidedly unclear point remains. Are the out-of-hospital births that the authors evaluated planned home births or incidental births of women who did not reach the hospital in time? Obviously the answer to this question has enormous influence on the analysis of the results. Although I am personally decidedly against home birth, supported in this by extensive literature, I however believe that the occurrence of hypothermia in an organized home birth is much more serious than in an accidental birth, for example on board an ambulance which urgently transfers a pregnant woman about to give birth. Reading the article all this is completely overlooked. Second point. The authors claim that the area included in their study is one with the lowest population density in the UK, if I understand correctly. Should I therefore assume that their reported data is even broader than I imagine? Third point. The authors state: experiences of the team varied and represent ethnic, cultural and geographical diversity. But what does
---

	this mean? A healthcare team, of whatever origin, must be experts in the subject it faces. Who could possibly care about the ethnicity of individual team members? Frankly this nonsense should not belong in an objective scientific article. Apart from these considerations, I have a basic doubt that I cannot place in the context of the article. In the UK, I know for a fact, the Neonatal Emergency Transport Service has a long and brilliant history of efficiency and excellent capacity. But who transported these children? Which staff assisted them? Staff with specific neonatal experience or general healthcare staff? There are many points that are not clear in this article. Another very difficult aspect to understand refers to the recruitment in phase B. But why did you think of paying the participants (25 pounds)? At best, this is a bias, if not worse. Then, the five questions used to identify as affecting neonatal temperature management during EMS calls. Even accepting the five questions, 4 and a half pages to address these five issues is unacceptable. This part of the text must be significantly reduced; perhaps with the help of a table. The article concludes that 35% of the children included in the study were hypothermic at the time of admission. This doesn't seem like a surprising result to me. The literature has widely established that being born outside an equipped hospital places the newborn at risk of many complications, including hypothermia. The authors state that in Finland the rate is 53% and that in France the rate is 56% of hypothermic children. So they should be happy with their result! That said, why don't they analyze their experience in comparison with the rest of Europe, trying to understand what is currently of a good level in their local organization and what could possibly need to be changed, offering suggestions to correct any shortcomings. Finally, I am not a native English speaker, so I do not express any judgment in relation to grammatical style, as I am not capable of doing so.
--	---

REVIEWER	Sabsabi , Bayane McGill University
REVIEW RETURNED	15-Dec-2023

GENERAL COMMENTS	I read the paper by Goodwin et al. It seems to me the article is in the scope of BMJ open. The authors describe the inequalities in birth before arrival and use mixed methods (quantitative and qualitative approach) to better answer the research question. In my opinion, the article is overall well organized and contains few grammatical and spelling errors. The tables and figures could perhaps be more refined in regards to the information presentation (for example include # of patients in the flowchart), and provide a describe in the figure 1 box. Its content is original and novel to the Birth before arrival literature; however, the research question is broad and perhaps the use of mixed methods aims to capture the wide range possible answers. I believe focusing on one methodology qualitative for instance and refining the research questions as to for instance, process, vs. experience, may tailor the discussion further. Moreover, the background did little to explain the the reason certain characteristics were chosen over others. Moreover, the type of statistical analysis needed to make this comparison was not done. Finally, the discussion lacks appropriate literature review and further
---

	discussion into the limitations of this type of study. In summary, the paper contains good elements to be published and would be asset to the neonatal BBA literature but it requires major revisions (in regard to statistical analyses and discussion) and requires further reconsiderations in its tables and figures and its discussion (thorough literature review and further descriptions around the cases risk factors). I do not recommend its application as it is presented, but after improvement it can be resubmitted for possible publication.
--	---

REVIEWER	Mistry, Aarti University of Nottingham
REVIEW RETURNED	25-Dec-2023

GENERAL COMMENTS	Overall: The key strengths and novelty of this paper comes from the thematic analysis conducted on the ambulance call handler calls and the qualitative method of utilising multi-displinary focus groups and PPI to review and discuss key themes. This method is clearly documented and represented in simple flowchart, however it would be useful to know how many call handling transcripts were analysed initially by the research team to formulate the themes and the screening processes they adopted (something potentially provided in a supplementary file) . I feel greater attention and discussions of the results from Phase B study is needed and should be considered more in when making your final conclusions. Particularly, highlighting future work, impact on local policy, guidelines, call handling alorightms etc. The primary objective (Phase A), doesnt really share any associated link with the Phase B part of the study. Furthermore, the Phase A objective requires redefining and needs to be more specific. In particular you mention about looking at the association of neonatal hypothermia with "poor outcomes" what specific outcome are referring to or how do you define this. It is evident that small sample size is some what inadequate in addressing this primary objective and unable to make accurate inferences around maternal inequalities around BBA and neonatal hypothermia. I note that this has been mentioned as part of your limitations. Results: Phase A results, could you combine Table 1 and Table 3, ie looking at mother and infant characteristic based on WHO definitions of hypothermia. As the catergory of hypothermia the patient falls in will reflect a different management approach. In Table 2, did you consider using social deprivation scores? In reference to Table 3, it would be useful to understand how you have defined prematurity, or provide a gestational age range for each group. You also discuss that there was greater BBA in mothers that had delayed or were late bookers (Table 2) , however this characteristic is not considered in Table 3. You infer that infants who are preterm with BBA were more prevalent to be hypothermic, this is not unexpected as there is strong evidence of being born preterm being a risk factor for hypothermia and hypoglycaemia. This is something that should be discussed in your limitations, as you cant adjust for these factors.
--

	When describing the outcomes of those infants who presented with hypothermia, more detail is required on the infants characteristics. For example, were the infants preterm, did they present with sepsis, which WHO category for hypothermia did they fall in ie moderate or severe hypothermia. Providing this context to your claims may provide reasons to why they had prolonged length of stay, and required greater interventions for their hypothermia management. Furthermore, you mention infants who were hypothermic were likely to need admission into NICU, are you referring to tertiary level neonatal care or are you referring to all types of neonatal units ie LNU and SCBU. Also what is meant by "advance care" this is a very vague term. It maybe useful to include a list of how you define different maternal and infant characteristics included in your tables as a supplementary file to this study.
--	---

VERSION 1 – AUTHOR RESPONSE

Reviewer 1 comments

COMMENT: First look. The authors state that in the UK, approximately 3,700 births per year occur outside of hospital. Frankly, that seems like a huge number to me. I admit that I know approximately the birth data in the UK, but I would say that the percentage of births outside hospital is therefore around 1.2-1.3% (Wales, Scotland, England) approximately the same as Iceland and Switzerland. In Europe is decidedly below 1% and even in Italy we are at 0.05-0.06%.

RESPONSE: Literature suggests that 0.5% of births in the UK are unplanned out of hospital births (Loughney et al – reference 1 in the paper). Using data from the Office for National Statistics on the number of births in the UK each year, 3,700 is roughly 0.5%. We have reason to believe that it is actually higher than this, however, as our local ambulance service alone attends nearly 600 of these births a year and other ambulance services have reported similar (and higher) numbers. We have added references to support this estimate.

LOCATION OF CHANGE: Background: Page 3

COMMENT: Are the out-of-hospital births that the authors evaluated planned home births or incidental births of women who did not reach the hospital in time?

Obviously the answer to this question has enormous influence on the analysis of the results. Although I am personally decidedly against home birth, supported in this by extensive literature, I however believe that the occurrence of hypothermia in an organized home birth is much more serious than in an accidental birth, for example on board an ambulance which urgently transfers a pregnant woman about to give birth. Reading the article all this is completely overlooked.

RESPONSE: This includes all births transported to hospital by the ambulance service 'Records were included if they related to a live birth in the prehospital setting where the baby was brought to hospital by EMS' – as per page 4. We have clarified that this is all babies brought in by the ambulance service. The majority will be unplanned out of hospital births (UOHB; including those who did not reach the hospital in time) but a minority may be planned home births where EMS skills/urgent transport is needed.

LOCATION OF CHANGE: Methods: Page 4

COMMENT: The authors claim that the area included in their study is one with the lowest population density in the UK, if I understand correctly. Should I therefore assume that their reported data is even broader than I imagine?

RESPONSE: Yes our study took place in an area of low population density. It is possible that the

issue of neonatal hypothermia is even more of a problem in higher density populations but we don't have the data to support this. We have added a sentence to the discussion to highlight that further work should replicate our hospital audit to confirm or refute this.

LOCATION OF CHANGE: Discussion: Page 15

COMMENT: The authors state: experiences of the team varied and represent ethnic, cultural and geographical diversity. But what does this mean? A healthcare team, of whatever origin, must be experts in the subject it faces. Who could possibly care about the ethnicity of individual team members? Frankly this nonsense should not belong in an objective scientific article.

RESPONSE: This sentence is taken from the section on Patient and Public Involvement (PPI) members. Equality, diversity and inclusion is a key part of UK standards for Public Involvement (<https://sites.google.com/nih.ac.uk/pi-standards/standards/inclusive-opportunities>), and as such, should be commented upon for transparency. The importance of diversity and inclusion also applies to research participants (staff or patients). If we do not include participants from diverse demographic backgrounds, then we cannot claim to be representing the thoughts/views of a diverse population. This could potentially lead to perpetuating health inequalities

(<https://evidence.nih.ac.uk/collection/why-we-need-more-inclusive-research>).

LOCATION OF CHANGE: N/A

COMMENT: The authors state: experiences of the team varied and represent ethnic, cultural and geographical diversity. But what does this mean? A healthcare team, of whatever origin, must be experts in the subject it faces. Who could possibly care about the ethnicity of individual team members? Frankly this nonsense should not belong in an objective scientific article.

Apart from these considerations, I have a basic doubt that I cannot place in the context of the article. In the UK, I know for a fact, the Neonatal Emergency Transport Service has a long and brilliant history of efficiency and excellent capacity. But who transported these children? Which staff assisted them? Staff with specific neonatal experience or general healthcare staff?

RESPONSE: The Neonatal Emergency Transport Service does not usually attend prehospital births in the UK. Instead, they transport neonates between healthcare units. Unplanned out of hospital births in the UK are attended by the Ambulance Service, and crews can be a mix of paramedics, emergency care assistants and ambulance technicians. They do get some training on maternity and neonatal care, but this is very limited. We have clarified this within the background section.

LOCATION OF CHANGE: Background: Page 3

COMMENT: Another very difficult aspect to understand refers to the recruitment in phase B. But why did you think of paying the participants (25 pounds)? At best, this is a bias, if not worse.

RESPONSE: Reimbursing participants and Patient and Public Involvement members for their time in research is standard practice in the UK, and guidance from INVOLVE sets the rate at £25 per hour (<https://www.nih.ac.uk/documents/payment-guidance-for-researchers-and-professionals/27392>). The ethics committee agreed that it is completely reasonable to reimburse participants for their time using these vouchers (HAS.22.01.061).

LOCATION OF CHANGE: N/A

COMMENT: Then, the five questions used to identify as affecting neonatal temperature management during EMS calls. Even accepting the five questions, 4 and a half pages to address these five issues is unacceptable. This part of the text must be significantly reduced; perhaps with the help of a table.

RESPONSE: The five themes summarise the findings of the qualitative phase of the research. Quotes have been placed into a table to shorten this part of the manuscript.

LOCATION OF CHANGE: Table 4

COMMENT: The article concludes that 35% of the children included in the study were hypothermic at the time of admission. This doesn't seem like a surprising result to me. The literature has widely

established that being born outside an equipped hospital places the newborn at risk of many complications, including hypothermia. The authors state that in Finland the rate is 53% and that in France the rate is 56% of hypothermic children. So they should be happy with their result! That said, why don't they analyze their experience in comparison with the rest of Europe, trying to understand what is currently of a good level in their local organization and what could possibly need to be changed, offering suggestions to correct any shortcomings.

RESPONSE: We agree that this is better than rates of neonatal hypothermia suggested in other countries, however we feel that 35% is still a significant proportion of babies arriving at hospital hypothermic, especially given the potential for poor outcomes. It is hard to compare data across EMS systems from different countries, as they operate differently, have different demographics and inequalities, and different protocols for responding to prehospital birth. However we have highlighted some potential areas for improvement in the UK, based on previous literature, and have added a sentence to the discussion to emphasise this point.

LOCATION OF CHANGE: Discussion: Page 15

COMMENT: Finally, I am not a native English speaker, so I do not express any judgment in relation to grammatical style, as I am not capable of doing so.

RESPONSE: N/A

LOCATION OF CHANGE: N/A

Reviewer 2 comments

COMMENT: The tables and figures could perhaps be more refined in regards to the information presentation (for example include # of patients in the flowchart), and provide a describe in the figure 1 box.

RESPONSE: We have added the number of participants in each set of focus groups, as well as the number of transcripts analysed. We have also edited the figure legend to add more description.

LOCATION OF CHANGE: Figure 1

COMMENT: Its content is original and novel to the Birth before arrival literature; however, the research question is broad and perhaps the use of mixed methods aims to capture the wide range possible answers. I believe focusing on one methodology qualitative for instance and refining the research questions as to for instance, process, vs. experience, may tailor the discussion further.

RESPONSE: Thank you for your comment. We considered splitting the two phases of the research into two separate papers, however the Journal Editor advised us not to do so, and felt it was better presented together.

LOCATION OF CHANGE: N/A

COMMENT: Moreover, the background did little to explain the reason certain characteristics were chosen over others.

RESPONSE: We included all the demographic characteristics that we were able to get from the Hospital Data. We have added a sentence in the Methods section to clarify this.

LOCATION OF CHANGE: Methods: Page 5

COMMENT: Moreover, the type of statistical analysis needed to make this comparison was not done.

RESPONSE: Unfortunately, we did not have large enough numbers to perform meaningful statistical analysis, so basic descriptive statistics were used on the advice of the statistics expert co-author. A sentence has been added to the methods section to clarify this, and we have also noted this in our limitations (Discussion). We have added a further sentence to the discussion to suggest that further work (with larger numbers) should be done to replicate and expand on this analysis.

LOCATION OF CHANGE: Methods: Page 5, and Discussion: Page 15

COMMENT: Finally, the discussion lacks appropriate literature review and further discussion into the

limitations of this type of study.

RESPONSE: We have expanded our discussion to include further literature review and limitations.

LOCATION OF CHANGE: Discussion: Pages 15-16

COMMENT: In summary, the paper contains good elements to be published and would be asset to the neonatal BBA literature but it requires major revisions (in regard to statistical analyses and discussion) and requires further reconsiderations in its tables and figures and its discussion (thorough literature review and further descriptions around the cases risk factors).

RESPONSE: Thank you – we have addressed your comments as above, and hope that you now find it suitable for publication.

LOCATION OF CHANGE: N/A

Reviewer 3 comments

COMMENT: The key strengths and novelty of this paper comes from the thematic analysis conducted on the ambulance call handler calls and the qualitative method of utilising multi-disciplinary focus groups and PPI to review and discuss key themes. This method is clearly documented and represented in simple flowchart, however it would be useful to know how many call handling transcripts were analysed initially by the research team to formulate the themes and the screening processes they adopted (something potentially provided in a supplementary file)

RESPONSE: Thank you for your comment. We have added this information to the flowchart (Figure 1) and methods section.

LOCATION OF CHANGE: Figure 1, and Methods: Page 5

COMMENT: I feel greater attention and discussions of the results from Phase B study is needed and should be considered more in when making your final conclusions. Particularly, highlighting future work, impact on local policy, guidelines, call handling algorithms etc.

RESPONSE: Thank you for your comment. We have expanded further on the discussion of Phase B of the study.

LOCATION OF CHANGE: Discussion: Page 16

COMMENT: The primary objective (Phase A), doesnt really share any associated link with the Phase B part of the study.

RESPONSE: We considered splitting the two phases of the research into two separate papers, however the Journal Editor advised us not to do so, and felt it was better presented together. The two phases of this work were funded as one project and are linked through a focus on inequalities and accessibility of advice. Through this work (as a whole) we wanted to ensure sure that we are aware of populations who might be most at risk of BBA/hypothermia following BBA, and are giving them accessible and easily understandable information. Through this, we hope to disproportionately improve outcomes for minoritized groups. We have added information to the background and methods sections to clarify this.

LOCATION OF CHANGE: Background: Page 3, and Methods: Page 4

COMMENT: Furthermore, the Phase A objective requires redefining and needs to be more specific. In particular you mention about looking at the association of neonatal hypothermia with "poor outcomes" what specific outcome are referring to or how do you define this. It is evident that small sample size is some what inadequate in addressing this primary objective and unable to make accurate inferences around maternal inequalities around BBA and neonatal hypothermia. I note that this has been mentioned as part of your limitations.

RESPONSE: We have expanded on our definition of poor outcomes in the objectives part of the methods section. As you mention, our sample size limited our ability to answer this objective fully, although it gives us some idea of those babies who may be more at risk of being hypothermic on arrival at hospital. Further work is needed to replicate and expand this work to explore areas for

targeted intervention. This has been added/clarified in the discussion.

LOCATION OF CHANGE: Methods: Page 4, Discussion: Page 15

COMMENT: Phase A results, could you combine Table 1 and Table 3, ie looking at mother and infant characteristic based on WHO definitions of hypothermia. As the category of hypothermia the patient falls in will reflect a different management approach.

RESPONSE: Thank you for your suggestion. We had a look at how we might be able to present Table 3 based on the WHO definitions of hypothermia, as you suggested. However, our statistics expert felt that the numbers were insufficient to categorise Table 3 in this way, as they could be misleading. We only had 2 babies with a temperature of $< 32^{\circ}\text{C}$ so this wouldn't have been useful as a column on its own. With only $n=2$ in a column, one mother/baby in a category accounts for 50% and so is not very helpful to the reader.

LOCATION OF CHANGE: N/A

COMMENT: In Table 2, did you consider using social deprivation scores?

RESPONSE: We were hoping to use social deprivation scores to explore inequalities based on this characteristic, however we were unable to get these scores from the data we were given by the Trusts (first half of postcode only). We have added a sentence on this to the Results section.

LOCATION OF CHANGE: Results: Page 7

COMMENT: In reference to Table 3, it would be useful to understand how you have defined prematurity, or provide a gestational age range for each group.

RESPONSE: Thank you for this comment. We have now re-classified prematurity to match World Health Organization classifications (Pre-term as <37 weeks; Term as $37+$ weeks) and defined this within our table. This has changed our numbers slightly (which have been updated), but does not change the finding that hypothermia on arrival was more prevalent for pre-term babies.

LOCATION OF CHANGE: Table 3

COMMENT: You also discuss that there was greater BBA in mothers that had delayed or were late bookers (Table 2) , however this characteristic is not considered in Table 3.

RESPONSE: We have added this information to the table, as requested.

LOCATION OF CHANGE: Table 3

COMMENT: You infer that infants who are preterm with BBA were more prevalent to be hypothermic, this is not unexpected as there is strong evidence of being born preterm being a risk factor for hypothermia and hypoglycaemia. This is something that should be discussed in your limitations, as you cant adjust for these factors.

RESPONSE: We have added this point to the discussion section, as suggested.

LOCATION OF CHANGE: Discussion: Page 15

COMMENT: When describing the outcomes of those infants who presented with hypothermia, more detail is required on the infants characteristics. For example, were the infants preterm, did they present with sepsis, which WHO category for hypothermia did they fall in i.e. moderate or severe hypothermia. Providing this context to your claims may provide reasons to why they had prolonged length of stay, and required greater interventions for their hypothermia management

RESPONSE: We have added this information to the results section, as suggested.

LOCATION OF CHANGE: Results: Page 9

COMMENT: Furthermore, you mention infants who were hypothermic were likely to need admission into NICU, are you referring to tertiary level neonatal care or are you referring to all types of neonatal units ie LNU and SCBU. Also what is meant by "advance care" this is a very vague term.

RESPONSE: We are referring to all types of neonatal units. This has been amended. We have

defined the term 'advanced care' as any care needed in addition to 'normal newborn care', for example the provision of oxygen or intravenous (IV) medication. This is from the BAPM Framework and has now been defined and cited within the text.

LOCATION OF CHANGE: Results: Page 9

COMMENT: It maybe useful to include a list of how you define different maternal and infant characteristics included in your tables as a supplementary file to this study.

RESPONSE: We have added a supplemental file for these definitions, as requested.

LOCATION OF CHANGE: Supplemental file 1

VERSION 2 – REVIEW

REVIEWER	Bellini, Carlo Giannina Gaslini Institute
REVIEW RETURNED	19-Feb-2024

GENERAL COMMENTS	I read with great attention the answers that the authors gave to the questions I had asked regarding their manuscript. First of all, I must admit that some questions I asked stemmed from my lack of knowledge of local regulations, such as the rules relating to interviews. The authors explained these points very well in their answers. Regarding my observations relating to substantive issues, the authors have received and argued convincingly and have modified the original text where necessary. Personally, I am satisfied with the revision work carried out by the authors and, obviously having heard the binding opinion of the Editor in Chief, I believe that the article is worthy of publication in its current version.
--

REVIEWER	Mistry, Aarti University of Nottingham
REVIEW RETURNED	18-Feb-2024

GENERAL COMMENTS	This draft of your manuscript is much improved from previous version. You have made good efforts to address the reviewers comments from the first review. The strength of your paper really comes from the thematic analysis and focus group work around EMS call handlers call. I have recommended need for minor revisions, my suggestions are below:  1. Background, you mention "Currently there is no published data to show the percentage of babies who are hypothermia on arrival to hospital following a BBA and the resulting outcomes of these babies" In the discussion you refer to study from France that shows % of BBA arriving hypothermic has increase NICU admission and length of stay. Can you reconsider this statement/ or be specific to the outcomes you are referring to. 2. Background: In reference to the 3rd paragraph, you refer to poor outcomes following BBA, what outcomes are you referring to? Or are you referring to the poor outcomes associated with neonatal hypothermia. Please provide clarity on this first statement and references as needed. 3. Table 3, Relabel your groups Hypothermia <36.5 and
--

	normothermia 36.5- 37.5 , as currently temp >36.5 could also include babies that present with hyperthermia 4. Table 3, remove ethnicity from the table as you already describe this in the text. 5. Table 3, Mother age at booking- can you present this as median (IQR) 6. Discussion: you refer "optimal equipment" do you mean thermoregulation equipment ie transwarmers. 7. Discussion: Requires some information about the limitations associated with Phase B part of the study and PPI 8. Discussion: As your study period was between 2018- 2021, Did you consider the impact of pandemic restrictions to results observed in your study. 9. Discussion: Paragraphs 2 and 3, talk about baby factors and women factors identified in those that were hypothermic and those that more likely to BBA. Please review these two paragraphs consider the implications of your findings . For example the other infant risk factors identified can you offer reasons to towards why these infants were more likely to be hypothermic. Does your study show that the lack of interventions by paramedics/parents meant these babies arrived more hypothermic or equipment.
--	--

VERSION 2 – AUTHOR RESPONSE

Thank you for your comments. Please see the responses in the table.

Reviewer 3		
1	Background, you mention "Currently there is no published data to show the percentage of babies who are hypothermia on arrival to hospital following a BBA and the resulting outcomes of these babies" In the discussion you refer to study from France that shows % of BBA arriving hypothermic has increase NICU admission and length of stay. Can you reconsider this statement/ or be specific to the outcomes you are referring to.	Thank you for highlighting this oversight. We have specifically indicated UK in the Background section.
2	Background: In reference to the 3rd paragraph, you refer to poor outcomes following BBA, what outcomes are you referring to? Or are you referring to the poor outcomes associated with neonatal hypothermia. Please provide clarity on this first statement and references as needed.	Thank you, we have added some more context to this first sentence of the third paragraph in the Background section.
3	Table 3, Relabel your groups Hypothermia <36.5 and normothermia 36.5- 37.5 , as currently temp >36.5 could also include babies that present with hyperthermia	We have added a line into Table 1 for hyperthermia, there was only one baby who had a temperature over 37.5. We have relabelled Table 3 as suggested and removed the 1 baby who was classed hypothermic from the analysis.

4	Table 3, remove ethnicity from the table as you already describe this in the text.	Thank you for this suggestion, we have removed the detail from the table regarding ethnicity.
5	Table 3, Mother age at booking- can you present this as median (IQR)	Thank you for this suggestion. We have amended this table as suggested.
6	Discussion: you refer "optimal equipment" do you mean thermoregulation equipment ie transwarmers.	We have provided more detail here - "such as transwarmers and accurate temperature monitoring devices".
7	Discussion: Requires some information about the limitations associated with Phase B part of the study and PPI	Thank you for highlighting this point. We have added a bullet point regarding the limitations of the focus groups to the section, 'Strengths And Limitations Of This Study' on Page 2. There is already a PPI section included at the bottom of Page 15.
8	Discussion: As your study period was between 2018- 2021, Did you consider the impact of pandemic restrictions to results observed in your study.	Thank you for highlighting this point. We have added a sentence to the limitations section on page 2 to address this.
9	Discussion: Paragraphs 2 and 3, talk about baby factors and women factors identified in those that were hypothermic and those that more likely to BBA. Please review these two paragraphs consider the implications of your findings . For example the other infant risk factors identified can you offer reasons to towards why these infants were more likely to be hypothermic. Does your study show that the lack of interventions by paramedics/parents meant these babies arrived more hypothermic or equipment.	We have added some more context into paragraphs 2 and 3 of the discussion section in response to this comment. From our study we do not know exactly when the temperature of the neonates dropped and whether it was due to a lack of parent, or clinician intervention. However we can surmise that leaving a baby on a cold

		floor due to call-handler advice will reduce neonatal temperature.
--	--	--